# Peritoneal dialysis modality transition and impact on phosphate and potassium serum levels

**Daniela Peruzzo[1], Murilo Guedes[1], John W. Larkin[1,2], Guilherme Yokoyama[1], Taynara Lopes dos Santos[1], Roberto Pecoits-Filho[1], Silvia Carreira Ribeiro[1], Alfonso Ramos[3], Pasqual Barretti[4]\*, Thyago Proença de Moraes[1], on behalf of the BRAZPD Investigators[¶]**

**1** Pontifícia Universidade Católica do Paraná (PUCPR), Curitiba, Brazil, **2** Fresenius Medical Care, Global Medical Office, Waltham, MA, United States of America, **3** Baxter Healthcare, Mexico City, Mexico, **4** Universidade Estadual Paulista (UNESP), Botucatu, Brazil

¶ Membership of the BRAZPD Investigators are listed in the Acknowledgments.
\* pbarretti@uol.com.br

**Data Availability Statement:** All relevant data are within the manuscript and its Supporting Information files.

## Abstract

Peritoneal dialysis (PD) modalities affect solute removal differently. However, the impacts of switching PD modalities on serum levels of biomarkers of different sizes are not known. Our objective was to analyze whether a change in the PD modality associates with the levels of two routine biochemical laboratories. In this multicentric prospective cohort study. we selected all patients who remained on a PD modality for at least 6 months and switched PD modality. Patients were also required to be treated with the same PD modality for at least 3 months before and after the modality change. The primary outcome was change in potassium and phosphate serum levels. We identified 737 eligible patients who switched their PD modality during the study. We found mean serum phosphate levels increased during the 3 months after switching from CAPD to APD and conversely decreased after switching to from APD to CAPD. In contrast, for potassium the difference in the mean serum levels was comparable between groups switching from CAPD to APD, and vice versa. In conclusion, CAPD seems to be as efficient as APD for the control of potassium serum levels, but more effective for the control of phosphate serum levels. The effect of a higher removal of middle size molecules as result of PD modalities in terms of clinical and patient-reported outcomes should be further explored.

## Introduction

Peritoneal dialysis (PD) is a lifesaving end-stage kidney disease (ESKD) treatment used for over forty years [1]. PD assumes some functions of the diseased kidney by removing toxic wasting retention solute products generated daily by metabolism. The removal of uremic retention solutes varies significantly in PD and depends on factors such as peritoneal membrane characteristics, dialysis prescription, dwell-time, fill volume, and total daily volume of PD exchanges [2–5].

**Funding:** The funders (Baxter Healthcare and Fresenius Medical Care) support in the form of salaries for authors [AR and JWL, respectively], but did not have any additional role in the study design, data collection and analysis, decision to publish, or preparation of the manuscript. The specific roles of these authors are articulated in the 'author contributions' section.

**Competing interests:** 'AR is employed by Baxter Healthcare. MG, JWL, TLS and DP are students at Pontifícia Universidade Católica do Paraná. JWL is employed by Fresenius Medical Care. GY, PB and SCR have no conflict to declare. TPM, RPF are employed by Pontifícia Universidade Católica do Paraná and are recipients of scholarships from the Brazilian Council for Research (CNPq), received consulting fees from Astra Zeneca and Baxter Healthcare, and speaker honoraria from Astra Zeneca, Lilly-Boehringer, Baxter and Takeda. RPF is employed by Arbor Research Collaborative for Health, and receives research grants, consulting fees, and honoraria from Astra Zeneca, Novo Nordisc, Akebia, and Fresenius Medical Care. This does not alter our adherence to PLOS ONE policies on sharing data and materials.

Mathematical models estimating peritoneal membrane transport suggest solute size is an important factor for dialysis clearance [3]. Specifically, middle molecular weight molecules demand more time of contact between PD solution fluid and the peritoneum to achieve target peritoneal clearance. In this context, continuous ambulatory peritoneal dialysis (CAPD) may favor the removal of larger molecules because of inherent characteristics of a longer dwell time compared to automated peritoneal dialysis (APD). Supporting this concept, a retrospective study of 371 patients from United Kingdom found phosphate removal was higher in CAPD compared to APD [6]. Although phosphate is only 96 Daltons, the phosphate molecule is surrounded by an aqueous cover and it behaves as a middle size molecule [6]. Questions still remain as to whether the higher clearance of phosphate in CAPD actually impacts serum phosphate levels yielding better phosphate control that stabilizes more patients in the recommended target levels.

In contrast, small molecular weight molecules such as potassium (39 Daltons) achieve equilibration with less dwell time during PD. This has been confirmed in a randomized cross-over study of 15 Belgium patients [7]. The dialytic mass removal of urea (60 Daltons) compared to other middle molecules was greater in APD. This finding was expected, provided the total daily exchange volume in APD is larger than CAPD. As far as the authors' knowledge, the impact of switching PD modalities on serum phosphate and potassium levels has not been established. Therefore, we performed an analysis of a nationally representative cohort to evaluate the impact of PD modality short-term changes on both serum potassium and phosphate levels.

## Materials and methods

### Study population and design

This was a cohort study from the BRAZPD II, which was a large national prospective cohort study that included patients from 122 dialysis centers throughout Brazil. Clinical and laboratorial data were prospectively collected monthly from December 2004 to January 2011. The study design and general characteristics of the BRAZPD II cohort have been previously published [8]. The BRAZPD study protocol was approved PUCPR Research Ethics Committee under the registration number MS n˚25000.187284/2004-01, and countersigned by the local Ethics Committee of all participating centers. BRAZPD study was performed in adherence with the Declaration of Helsinki and all patients signed informed consent.

In this study, we used data from a select group of patients who remained on a PD modality for at least 6 months after initiating kidney replacement therapy (KRT) with PD. We included data from patients who had their PD modality switched from APD to CAPD, or vice versa. Included patients were required to be treated with the same PD modality for at least 3 months before and after the modality change. Our objective was to analyze whether a change in the PD modality associates with the levels of two routine biochemical laboratories, namely potassium (a small size molecule) and phosphate (a small size molecule that behaves as a middle size molecule). Data from eligible patients were stratified into two groups: Starting on CAPD, with patients starting on CAPD who changed to APD and Starting on APD, with patients starting on APD who changed to CAPD (Fig 1).

The presence of residual kidney function (RKF) was recorded at baseline (during the first 90 days of PD) using a single 24-hour urine measurement; the presence of RKF was defined as having >100 mL per day of urine output. The prescribed total daily exchange volume of was captured during concurrent periods with biochemical markers 3 months before and after the change in PD modality. To evaluate effect modification, subgroup analyses in patients with RKF data available was carried out.

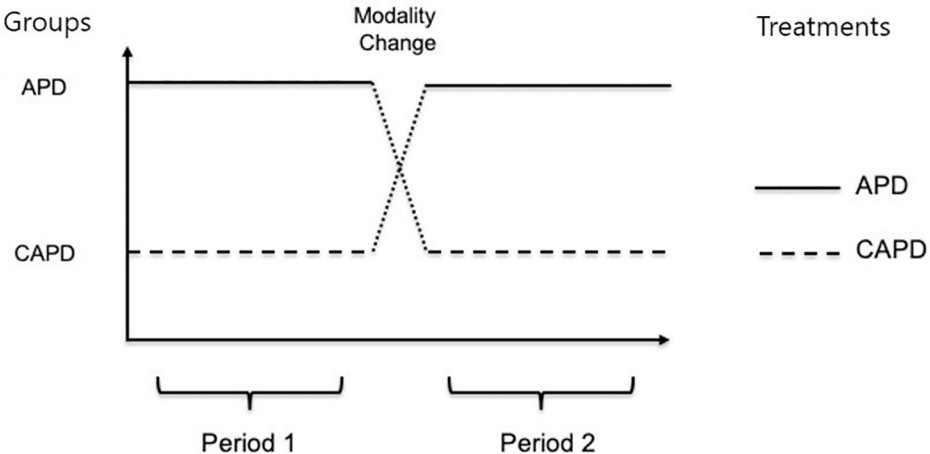

**Fig 1. Study design.** Periods 1 was 3 months before the modality change. Period 2 was 3 months after the modality change. Patients without sufficient laboratory data (less than 2 measurements) in both the before and after time periods were excluded from the study.

## Statistical analysis

Continuous variables were expressed as mean ± standard deviation (SD) or median and interquartile range. Categorical variables (e.g., gender, race, primary etiology of kidney disease, presence of comorbid conditions, initial KRT therapy, current PD modality) were expressed as frequencies or percentages.

As appropriate, χ2, $t$ test, or Wilcoxon methods were used to compare demographic and clinical characteristics between groups at baseline (first 90 days of PD). For assessment of the main outcome of the difference in serum potassium and phosphate levels before versus after switching PD modalities, we used two different statistical methods for comparisons. Fractional polynomials functions were estimated to provide the predicted and their 95% CI´s values using regression models as described by Royston and Altman in 1994. The first method used a non-parametric Wilcoxon test for repeated measures. The second method is a test developed to crossover studies and available in the software STATA. This test use analysis of variance to provide values for carryover, sequence, period, and intervention effects (*pkcross*). We performed a sensitivity analysis stratifying our study population according to the presence or not of residual renal function at baseline. Statistical significance was set at the level of $p < 0.05$. All analyses were performed using STATA 14.0.

## Results

Among 5707 PD patients who initiated PD and had at least 6 months of follow-up during the study, 22% (n = 1232) of patients had a change in their PD modality (Fig 2). The time to transference for all patients and according to the subgroup (CAPD to APD or APD to CAPD) are showed in a S1 Fig.

There were 495 patients who had a change in their PD modality but remained on the new modality for less than 3 months; these patients were excluded from the analysis. We identified 737 eligible patients who switched their PD modality during the study.

The baseline characteristics of the BRAZPD cohort and the select study populations are shown in Table 1. The characteristics of patients who had a change in their PD modality was not remarkably different compared to the overall cohort population. Given the study design,

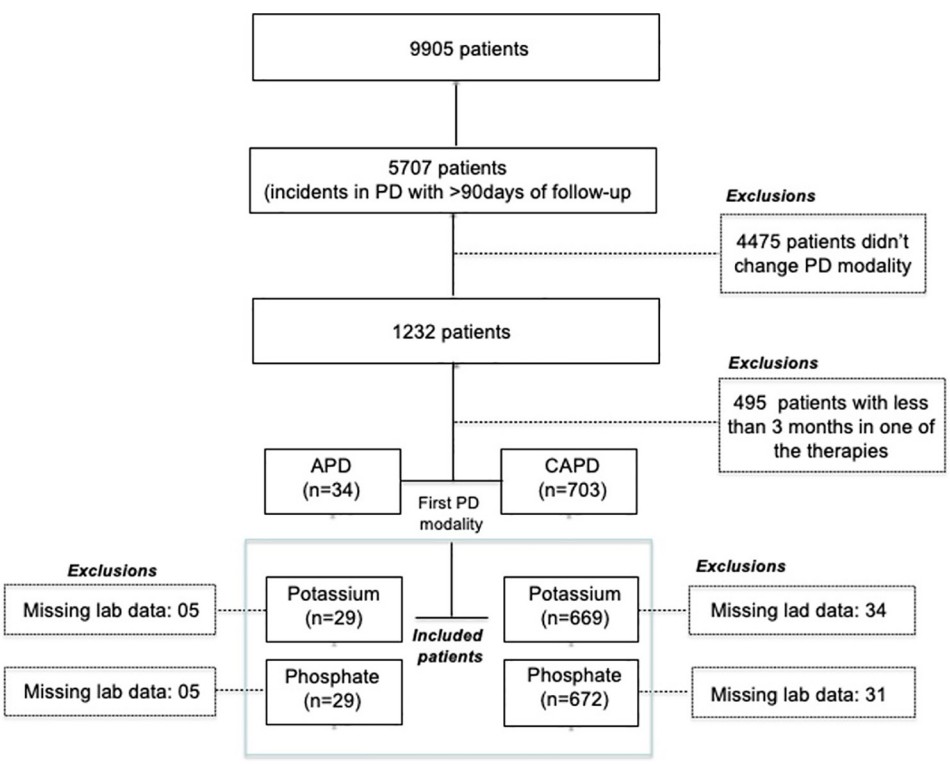

**Fig 2. Study flowchart.**

APD was used as the initial PD modality the in a lower proportion of patients included in this analysis compared to the overall cohort.

Mean phosphate levels were about 0.7 mg/dL higher in patients treated with APD compared to CAPD during the 3 months before a PD modality change (Table 2). We found mean serum phosphate levels increased during the 3 months after switching from CAPD to APD (Fig 3A), and conversely decreased after switching to from APD to CAPD (Fig 3B).

**Table 1. Clinical and demographic characteristics.**

| Variables | Study population | Original cohort |
|---|---|---|
| Age | 58.4±15.3 | 59.4±16.0 |
| Body mass index | 25.5±4.9 | 24.7±4.7 |
| Coronary artery disease | 17% | 21% |
| Diabetes | 49% | 44% |
| Gender (male) | 49% | 48% |
| Hypertension (yes) | 76% | 73% |
| Initial modality (APD) | 5% | 46% |
| Literacy (> 4 years formal study) | 41% | 35% |
| Peripheral artery disease (yes) | 20% | 21% |
| Pre-dialysis care (yes) | 59% | 51% |
| Previous hemodialysis (yes) | 42% | 36% |
| Race (White) | 62% | 64% |
| Residual renal function (yes) | 76% | 65% |

**Table 2. Impacts of PD modality switch on serum phosphate and potassium levels.**

| Outcome | Mean values before switching | | Mean change at 3 months | | pkcross results (p value) | | |
|---|---|---|---|---|---|---|---|
| | Sequence | | Sequence | | | | |
| | Starting on APD | Starting on CAPD | Starting on APD | Starting on CAPD | Intervention* | Sequence** | Period*** |
| Phosphate (mg/dL) | 5.49±1.32 | 4.75±1.30 | -0.68±1.68 | 0.25±1.34 | 0.002 | 0.38 | 0.07 |
| Potassium (mEq/L) | 4.60±0.73 | 4.28±0.75 | -0.05±0.84 | 0.05±0.74 | 0.92 | 0.06 | 0.61 |
| *Patients with residual kidney function at baseline* | | | | | | | |
| Phosphate (mg/dL) | 5.14±0.58 | 4.68±1.23 | -0.36±1.25 | 0.29±1.27 | 0.02 | 0.48 | 0.03 |
| Potassium (mEq/L) | 4.67±0.84 | 4.24±0.73 | -0.01±0.91 | 0.08±0.75 | 0.95 | 0.04 | 0.96 |
| *Patients without residual kidney function at baseline* | | | | | | | |
| Phosphate (mg/dL) | 5.97±1.87 | 5.00± 1.50 | -1.13±2.13 | 0.11±1.55 | 0.86 | 0.78 | 0.32 |
| Potassium (mEq/L) | 4.52±0.58 | 4.39±0.78 | -0.10±0.78 | -0.05±0.72 | 0.10 | 0.78 | 0.68 |

*Intervention describes whether modality switching affect the serum biomarkers levels

**Sequence evaluate whether the order (APD or CAPD first) affect the results

*** Period compare the results of the first phase of the study versus the second without considering the PD modality.

Among patients starting on APD, the mean change in phosphate was -0.68±1.68 mg/dL during the 3 months after switching to CAPD and the prevalence of patients with hyperphosphatemia (phosphate >5.5 mg/dL) decreased from 41.3 to 29.3% (Fig 4). In the patients starting on CAPD, mean phosphate levels increased 0.25±1.34 mg/dL during the 3 months after switching to APD and the proportion of patients with hyperphosphatemia increased from 33.5 to 37.9% (Fig 4).

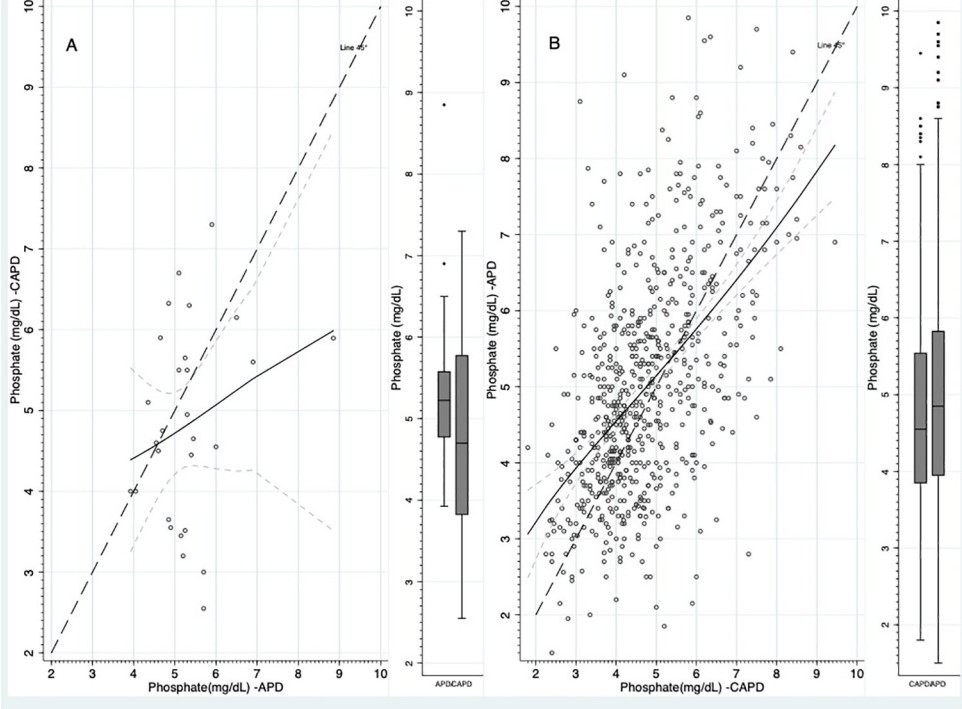

**Fig 3. Phosphate serum levels.** A: Starting on CAPD and switching to APD, B: Starting on APD and switching to CAPD. The continuous black line was predicted with fractional polynomials. The dashed gray lines are the 95% confidence intervals for continuous black line. The boxplots depict the mean value, interquartile range and min. max. values of phosphate during the time on CAPD and APD.

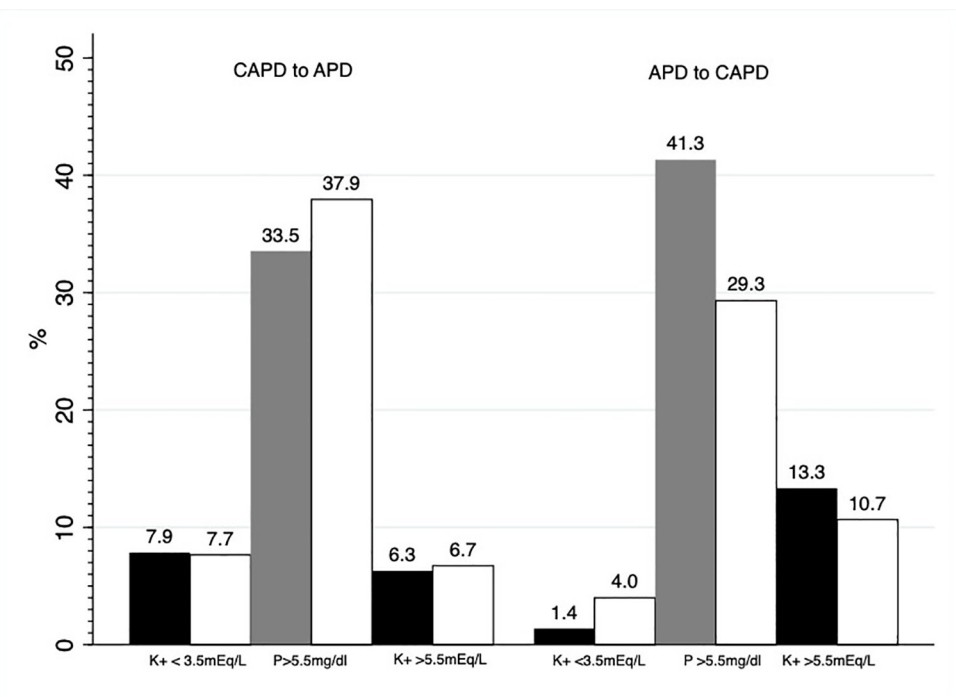

**Fig 4. Percentage of patients with hypokalemia (potassium <3.5 mEq/L), hyperkalemia (potassium >5.5 mEq/L) and hyperphosphatemia (phosphate >5.5 mg/dL) before (black/gray columns) and after (white columns) PD modality switch.**

In the sensitivity analysis that stratified the population according to baseline RKF, we found consistent findings in patients with RKF. However, there were no differences in mean phosphate levels before versus after switching from CAPD to APD, or vice versa.

The percentage of patients with a phosphate binder prescribed before and after switching for the group starting on CAPD was 64 and 67% respectively, and for the group starting on APD 63 and 60% respectively.

Mean serum potassium levels were consistent between patients treated with CAPD or APD during the 3 months before a PD modality change (Table 2). During the 3 months after patients changed their PD modality, the difference in the mean serum potassium levels was comparable between groups switching from CAPD to APD, and vice versa (Table 2; Fig 5).

In the sensitivity analysis that stratified the population according to baseline RKF, we found switching from CAPD to APD was associated with a minor crossover sequence association for a 0.08±0.75 mEq/L higher serum potassium level (p = 0.04) in patients with RKF, yet no significant crossover intervention or period associations were identified. The difference in the percentage of patients with hyperkalemia (potassium >5.5 mEq/L) before versus after switching a PD modality was minimal in patients who switched from CAPD to APD (difference: +0.4%), and small in patients who switched from APD to CAPD (difference: -2.6%) (Fig 5).

## Discussion/Conclusion

To the best of our knowledge, this is the largest study to examine the impacts of switching PD modalities on serum levels of two important biomarkers used in the management of chronic kidney disease, namely potassium and phosphate.

Potassium disorders are a life threating condition commonly found in later stages of chronic kidney disease. Hyperkalemia is frequently the trigger to initiate a kidney replacement

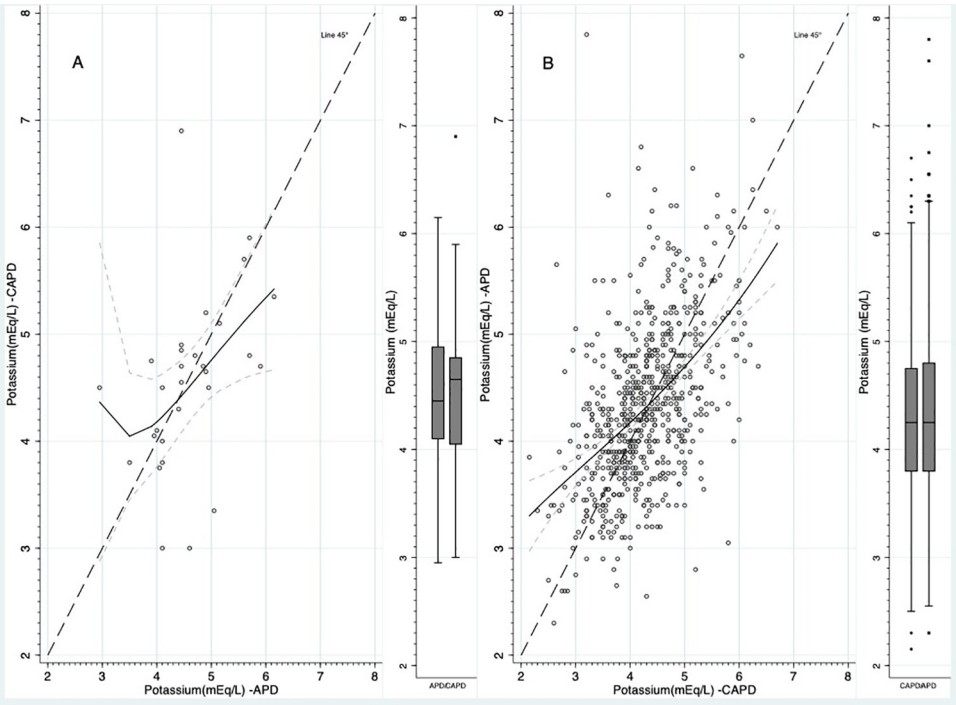

**Fig 5. Potassium serum levels.** A. Starting on CAPD and switching to APD; B Starting on APD and switching to CAPD. The continuous black line was predicted with fractional polynomials. The dashed gray lines are the 95% confidence interval for continuous black line. The boxplots depict the mean value, interquartile range and min. max. values of potassium during the time on CAPD and APD.

therapy and its prevalence after initiation of PD ranges from 5 to 10% [9–11]. When PD is initiated, the therapy can effectively control potassium levels and the process of removal occurs mainly by diffusion, given that PD solutions are characteristically free of potassium [12]. In contrast, the transport by convection is minimal: the estimated potassium removal with 1L of ultrafiltration range only from 3 to 7 mEq depending on the potassium serum concentration [12].

The balance between the blood and the dialysate happens quickly for small molecules as it is the case of potassium [13]. The typical short dwell time found with APD (around 1 to 2 hours) is normally enough to reach this balance and both CAPD and APD are considered equally effective therapies to remove potassium and avoid hyperkalemia. Our study confirms this hypothesis and shows the unremarkable impact of PD modality on potassium levels with a similar prevalence of hyperkalemia and a very small change in mean potassium when switching from CAPD to APD and vice-versa. We also evaluated the prevalence of hypokalemia, which is much more common in PD (up to 30% in some descriptions) [10, 14].

Hypokalemia in APD patients is a concern because the volume prescribed to APD is normally 50% higher compared to CAPD [5]. Such difference could potentially increase the risk of hypokalemia given that the balance for small molecules is reached within just a couple hours of a dwell. However, our findings are in contrast with this hypothesis, since the prevalence of patients with hypokalemia did not change with switching PD modality in the short-term. The 2.7% difference in the subgroup of patients who transitioned from APD to CAPD must be interpreted cautiously in light of the sample size of the group.

In contrast to small molecules, the transport of middle size molecules through the peritoneal membrane demands more time to achieve balance [12]. Phosphate is a good example to

demonstrate the behavior of middle size molecules removal in PD. Accordingly, removal of phosphate was greater in CAPD compared to APD in a British study with 87 individuals on chronic PD [6]. However, as reported by Badve and cols [4], the removal of phosphate through the peritoneal membrane depends on the membrane permeability, and as such, comparisons between CAPD and APD may be affected by peritoneal membrane characteristics. In this study, peritoneal phosphate clearances were similar between high transporters in either CCPD or APD, but was higher in CAPD individuals within the group defined as high-average transporters and even higher in the subgroup of low transporters (4). However, no study of our knowledge demonstrated in a large population whether this process could result in consistent differences in phosphate serum levels.

Our study clearly shows that phosphate levels increase when the patient is switched from CAPD to APD and decrease when switched from APD to CAPD. These results are in line with previous data and confirms the general concept that CAPD is more effective not only to removal middle size molecules, but also do it in a way that also changes phosphate serum levels. Despite our findings, a previous retrospective, single center study from Germany no differences were found between modalities [15]. In this article, D/P creatinine was evaluated but other significant confounders were not, as RKF for example. However, it is important to note that phosphate is a water-soluble molecule and that its behavior seems to be different from protein-binding molecules as p-cresol. This difference has been elegantly shown previously by Bammens et al and seems to be associated with uremic symptoms although in a small single center study [16]. Our sensitivity analysis observed consistent findings for increases in phosphate levels in patients with RKF who were switched from CAPD to APD, and vice versa. However, phosphate levels were not found to be different after a PD modality in patients without the presence of RKF. The influence of RKF appears to assist in the more efficient removal of phosphate in CAPD versus APD.

This study has important strengths, from the sample size to the design and robust statistical approach. Although BRAZPD does not have data on membrane profile, the combined results of phosphate reduction and the higher prevalence of patients on APD suggests that the original indication of APD is likely not based only on the individual membrane profile. Also, we explored the important potential effect modification of RKF in a priori sensitivity analysis. Also, our results are unlikely to suffer great impacts by confounders because the study design makes the patient his/her own control similar to a cross-over design.

Although there are many strengths, this study does have some limitations to take into account when reading interpreting the results, including: we cannot rule out confounding by indication in this analysis and we only analyzed short-term changes; we explored a potential association between peritonitis episodes and the modality conversion (no association was found) (S2 Fig). However, we cannot completely rule out the possibility that for some small group of patients the conversion could have occurred during the remission phase. We don't have data on whether the patient was on NIPD or CCPD; in addition, we are aware that when a patient is convert from one PD modality to another it is very difficult to define equivalence because many confounding factors are presented, particularly patient´s membrane profile and the dwell time; finally, we also did not have data on potassium and phosphate intake, as well as the doses of phosphate binders.

In conclusion, in this crossover study in a nationally representative cohort, CAPD seems to be as efficient as APD for the control of potassium serum levels, but more effective for the control of phosphate serum levels. The effect of a higher removal of middle size molecules as result of PD modalities in terms of clinical and patient-reported outcomes should be further explored.

## Supporting information

**S1 Fig. Time to change from continuous peritoneal ambulatory (CAPD) to automated peritoneal dialysis (APD) and from APD to CAPD.**
(JPG)

**S2 Fig. Time to peritoneal modality change and time of peritonitis.** All dots above the continuous black line represents peritonitis episodes that occurred after modality change. All dots below the dashed black line are peritonitis episodes that occurred at least 3 months before modality change.
(JPG)

## Acknowledgments

The authors thank the following centers which participated in BRAZPD and contributed with patient´s data used in this article: Ameneg, Associaçao Hosp Bauru, Biocor Hosp Doencas Cardiologicas, Casa De Saude E Mat N.Sra Perp Socorro, Cdr Curitiba, Cdr Goiania, Cdr Imperatriz, Cdr São Jose Pinhais, Cdtr_Centro Dialise Transplante Renal, Centro Nefrologia Teresopolis, Centro Nefrologico Minas Gerais, Centro Trat Doencas Renais Joinville, Centro Tratamento Renal Zona Sul, Clinef Rio De Janeiro, Clinepa Clinica De Nefrologia Da Paraiba, Clines, Clinese, Clinica Do Rim Do Carpina, Clinica Evangelico S/C Ltda, Clinica Nefrologia De Franca, Clinica Nefrologia Santa Rita, Clinica Nefrologica Sao Goncalo, Clinica Paulista Nefrologia, Clinica Renal Manaus, Clinica Senhor Do Bonfim, Clinica Senhor Do Bonfim Ltda Filial, Clinica Tratamento Renal, Cuiaba _Cenec, Clire Clinica Doencas Renais, Famesp Botucatu,Unicamp_Univ. Est Campinas, Hosp. Clinicas Fmrpusp, Fundaçao Civil Casa Mis Franca, Fundaçao Inst Mineiro Est Pesq Nefrol, Gamen Rio De Janeiro, Gdf Hospital De Base, Histocom Sociedade Civil Ltda, Hosp Univ Prof Edgard Santos, Hosp Benef. Portuguesa Pernambuco, Hospital Cidade Passo Fundo, Hospital Clinica Univ Fed Goias, Hospital E Maternidade Angelina Caron, Hospital Evangelico Vila Velha Es, Hospital Geral Bonsucesso, Hospital Geral de Goiania, Hospital Infantil Joana De Gusmão, Hospital São Joao Deus, Hospital São Jorge, Hospital São Jose Do Avai, Hosp Sao Vicente De Paula_J Pessoa, Hosp Sao Vicente De Paulo, Hospital Servidor Do Estado Ipase, Hospital Univ Presidente Dutra Ma, Hospital Universitario Antonio Pedro, Hospital Vita Volta Redonda S/A, Iamspe Sao Paulo, Imip, Inst Capixaba Doencas Renais, Inst Capixaba Doencas Renais Cariacica, Inst Capixaba Doencas Renais Serra, Inst Do Rim De Fortaleza, Inst Do Rim De Marilia, Inst Do Rim Do Parana S/C Ltda, Inst Do Rim Santo Antonio Da Platina, Inst Hemodialise De Sorocaba, Inst Medicina Nuclear Endocrina, Inst Nefrologia De Mogi Das Cruzes, Inst Nefrologia De Suzano, Inst Nefrologia Souza E Costa, Inst Urologia E Nefrol Barra Mansa, Inst Urologia E Nefrol Sj Rio Preto, Medservsp, Nefrocentro, Nefroclinica Caxias Do Sul, Nefroclinica Foz Do Iguacu, Nefroclinica Uberlandia, Nefron Clinica Natal, Nefron Contagem, Nephron Pelotas, Nephron São Paulo, Nucleo Nefrologia Belo Horizonte, Pro Nephron, Prorim Campos Dos Goitacaze, Puc Porto Alegre, Renalcare Serviços Medicos Ltda, Renalcor Angra Dos Reis, Renalcor Rio De Janeiro, Renalvida, Rien Rio De Janeiro, Santa Casa De Adamantina, Santa Casa De Jau_Unefro, Santa Casa De Marilia, Santa Casa De Ourinhos, Santa Casa De Santo Amaro, Santa Casa De Sao Jose Dos Campos, Santa Casa De Votuporanga, Serv De Nefrologia De Ribeirao Preto, Uerj_Hosp. Clin. Univ. Est. Rio De Janeiro, Uni Rim Joao Pessoa, Unidade Nefrologia Assis, Unirim Unidade De Doenças Renais, Unirim Unidade Renal Do Portao, Untr Unidade Nefrologia Transplante.

## Author Contributions

**Conceptualization:** Daniela Peruzzo, John W. Larkin, Taynara Lopes dos Santos, Roberto Pecoits-Filho, Silvia Carreira Ribeiro, Alfonso Ramos, Pasqual Barretti, Thyago Proença de Moraes.

**Data curation:** John W. Larkin, Taynara Lopes dos Santos, Roberto Pecoits-Filho, Alfonso Ramos, Pasqual Barretti, Thyago Proença de Moraes.

**Formal analysis:** Thyago Proença de Moraes.

**Funding acquisition:** Roberto Pecoits-Filho.

**Investigation:** Daniela Peruzzo.

**Project administration:** Thyago Proença de Moraes.

**Writing – original draft:** Daniela Peruzzo, Murilo Guedes, John W. Larkin, Guilherme Yokoyama, Taynara Lopes dos Santos, Roberto Pecoits-Filho, Alfonso Ramos, Pasqual Barretti, Thyago Proença de Moraes.

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
