## [Decision Letter · Decision Letter 0]

21 Apr 2021

PONE-D-21-02597

Peritoneal dialysis modality transition and impact on phosphate and potassium serum levels.

PLOS ONE

Dear Dr. Barretti,

Thank you for submitting your manuscript to PLOS ONE. After careful consideration, we feel that it has merit but does not fully meet PLOS ONE’s publication criteria as it currently stands. Therefore, we invite you to submit a revised version of the manuscript that addresses the points raised during the review process.

We look forward to receiving your revised manuscript.

Kind regards,

Gianpaolo Reboldi, MD, MSc, PhD

Academic Editor

PLOS ONE

Journal Requirements:

2. Please provide clinical trial registration information for the associated  BRAZPD II clinical trial.

[The BRAZPD cohort was funded by Baxter Healthcare, Brazil. The current data extraction and analysis was supported by an investigator driven study grant provided to the PontificiaUniversidade Católica do Paraná, as part of the Clinical Evidence Council Program from Baxter Healthcare.]

 [The funders had no role in study design, data collection and analysis, decision to publish, or preparation of the manuscript.]

[AR is employed by Baxter Healthcare.

MG, JWL, TLS and DP are students at Pontifícia Universidade Católica do Paraná.

JWL is employed by Fresenius Medical Care.

GY, PB and SCR have no conflict to declare

TPM, RPF are employed by Pontifícia Universidade Católica do Paraná and are recipients of scholarships from the Brazilian Council for Research (CNPq), received consulting fees from Astra Zeneca and Baxter Healthcare, and speaker honoraria from Astra Zeneca, Lilly-Boehringer, Baxter and Takeda.

RPF is employed by Arbor Research Collaborative for Health, and receives research grants, consulting fees, and honoraria from Astra Zeneca, Novo Nordisc, Akebia, and Fresenius Medical Care.].   

We note that one or more of the authors are employed by a commercial company: Fresenius Medical Care and Baxter Healthcare

5. Please amend the manuscript submission data (via Edit Submission) to include author Guilherme Yokoyama and Silvia Carreira Ribeiro.

6. One of the noted authors is a group or consortium [on behalf of the BRAZPD Investigators]. In addition to naming the author group, please list the individual authors and affiliations within this group in the acknowledgments section of your manuscript. Please also indicate clearly a lead author for this group along with a contact email address.

Reviewers' comments:

Reviewer's Responses to Questions

**Comments to the Author**

1. Is the manuscript technically sound, and do the data support the conclusions?

Reviewer #1: Partly

Reviewer #2: Partly

Reviewer #3: Partly

2. Has the statistical analysis been performed appropriately and rigorously? 

Reviewer #1: No

Reviewer #2: Yes

Reviewer #3: Yes

3. Have the authors made all data underlying the findings in their manuscript fully available?

Reviewer #1: No

Reviewer #2: Yes

Reviewer #3: Yes

4. Is the manuscript presented in an intelligible fashion and written in standard English?

Reviewer #1: Yes

Reviewer #2: Yes

Reviewer #3: Yes

5. Review Comments to the Author

Reviewer #1: This is a cross-over study looking at subjects that had a change in their PD modality. Participants were required to be on a steady modality for at least 3 months before study inclusion, and then at least 3 months after a shift in their modality. Measurements were taken pre and post shift.

The authors mention that all data is available either in the manuscript or the supplemental materials. There appear to be no supplemental materials, and the only data presented in the manuscript is means and standard deviations. This is not making all data publicly available. The authors should repose their raw data, or indicate why doing so is not feasible (includes protected health information).

Not enough information is provided in the statistical analysis section. Chi-square tests, t-tests, and Wilcoxon methods are mentioned as being used. However, it is not clear where any of these tests were run, as there do not appear to be any Confidence intervals or p-values reported from any of those tests. To be reproducible, the reader should be easily able to identify what type of test was run for what comparison, and be able to assess whether those results were significant or not. That is completely missing from this manuscript. The authors mention in the results/discussion sections that sensitivity analysis was performed. However, there is no mention of this being done in the statistical analysis section. The statistical analysis section should include mention of how the sensitivity analysis was done. Also, more information is needed about the analysis of the cross-over data. Simply stating what statistical software package was used for the analysis is insufficient. What statistical method(s) was used to analyze the data, and how was that analysis performed? That is what needs describing, in addition to including the package used to perform the analysis. Overall, more detail is needed.

Figures 3 and 5 need more explanation. What is the dashed black line? How was the fractional polynomial prediction performed? How were the 95% CI's obtained from that fit? This information should be provided in the statistical analysis section. Additionally, are the gray dashed lines for the solid black line or the dashed black line? What are the boxplots depicting? Mention of them is not included in the figure description. For Figure 5B, there is no mention of portion B in the figure description.

Reviewer #2: This is an interesting and well-written paper.

My major comments:

1. In the Abstract you state " In this multicentric prospective cohort study. we selected all patients who remained on a PD modality for at least 6 months and switched PD modality." whereas in the Introduction section you write " Therefore, we designed a cross-over analysis of a nationally representative cohort to evaluate the impact of PD modality short-term changes on both serum potassium and phosphate levels. ". I am confused. BRAZPD was a multicentric prospective observational cohort study, but this paper is actually a cross-over analysis evaluating the impact of PD modality short-term changes on both serum potassium and phosphate levels. It would be nice to make this distinction clear for the readers.

2. The Abstract and Final conclusion is "The effect of a higher removal of middle size molecules as result of PD modalities in terms of clinical and patient-reported outcomes should be further explored ". I do not fully agree with the conclusion and refer to the 2003 KI paper (Removal of middle molecules and protein-bound solutes by peritoneal dialysis and relation with uremic symptoms; VOLUME 64, ISSUE 6, P2238-2243, DECEMBER 01, 2003) by Bert Bammens, Pieter Evenepoel, Kristin Verbeke, Yves Vanrenterghem (Attached a pdf of the article). I do not fully agree with the conclusion as Bammens' paper already suggested that protein-bound solutes were involved in the pathophysiology of uremic symptoms. During peritoneal dialysis p-cresol behaves like β2m, probably due to its protein binding. The total clearance of both molecules is significantly lower as compared to water-soluble solutes and mainly depends on residual renal function. Your study may confirm earlier findings by Bammens et al. Do you agree ? I consider their paper an important reference to yours.

3. How did you consider a prescription equivalence in dialysis effectiveness when a patient was converted from CAPD to APD or vice-versa ? For example, did all CAPD patients on 4 exchanges with only 1.5% Dianeal PD solutions converted to 10 liters 1.5% Dianeal PD solution exchanges during the night and 2 liters 1.5 % Dianeal during the day (long dwell) ? Probably not, right ? Any changes in prescription may have affected the clearances; especially total daily infused volume, total daily dialysis time (CAPD usually 24 hours, but in APD you might have a "dry day" or even an extra exchange). This could be a limitation of the study.

4. It would be important to describe the reasons for conversion from CAPD to APD and vice-versa ? For example, in the case of peritonitis, were the analysis made after complete remission of the peritonitis ?

5. Did you analyse the effect of the time-window (vintage) from the start of PD (CAPD or APD) to the day the patient was "enrolled" in this specific study ? I can imagine that some occur within a short time-window and others within a long time-window.

6. In the Introduction section you very nicely describe that "Although phosphate is only 96 Daltons, the phosphate molecule is

surrounded by an aqueous cover and it behaves as a middle size molecule ". However, in the Material and Methods section, you describe " Our objective was to analyze whether a change in the PD modality associates with the levels of two routine biochemical laboratories, namely potassium (a small size molecule) and phosphate (a middle size molecule)".

I suggest a change in this last statement from ".......and phosphate (a middle size molecule)" to......and phosphate (a small size molecule that behaves as a middle size molecule)".

My minor comments:

1. In the affiliation of the authors in the first page, there must be some error for author Alfonso Ramos affiliation. It cannot be as it is now " Institution Alfonso, Mexico city, Mexico ". It is needed to be corrected.

2. BRAZPD has been fully supported/financed by Baxter Health Care and enrolling only Baxter PD patients in Brazil.. Therefore, it is a surprise to see a Fresenius employee being a co-author. Has Fresenius Medical Care given any financial support to this specific study ?

Reviewer #3: The question of phosphate elimination remains important in peritoneal dialysis patients since strong dietary phosphate restriction may lead to malnutrition and phosphate binder intake may increase pill burden and the risk of side effects. Moreover, changes in serum potassium levels may increase the risk for life threatening complications. This multicentric prospective cohort study aimed to examine the influence of a switch from CAPD to APD and vice versa on serum phosphate and potassium levels and the results may contribute to better clinical understanding.The potential influence of residual kidney function (RKF) was addressed as well as the limitations due to the lack of information on nutritional intake of phosphate and potassium and the specific doses of phosphate binders. Concerning the presentation of the key findings in Figure 3 A/B and 5 A/B the description of Figure 3 on page 13 and of Figure 5 on page 14 seems mismatching. It would be helpful if you could clarify this and/or add a more detailed description of Figure 3 and 5 as they are not self explaining especially with regard to the sample size.

6. PLOS authors have the option to publish the peer review history of their article (what does this mean?). If published, this will include your full peer review and any attached files.

Reviewer #1: No

Reviewer #2: **Yes: **Jose Divino

Reviewer #3: No

---

## [Decision Letter · Decision Letter 1]

25 Aug 2021

Peritoneal dialysis modality transition and impact on phosphate and potassium serum levels.

PONE-D-21-02597R1

Dear Dr. Barretti,

We’re pleased to inform you that your manuscript has been judged scientifically suitable for publication and will be formally accepted for publication once it meets all outstanding technical requirements.

Kind regards,

Gianpaolo Reboldi, MD, MSc, PhD

Academic Editor

PLOS ONE

Additional Editor Comments (optional):

Reviewers' comments:

Reviewer's Responses to Questions

**Comments to the Author**

1. If the authors have adequately addressed your comments raised in a previous round of review and you feel that this manuscript is now acceptable for publication, you may indicate that here to bypass the “Comments to the Author” section, enter your conflict of interest statement in the “Confidential to Editor” section, and submit your "Accept" recommendation.

Reviewer #1: All comments have been addressed

Reviewer #2: All comments have been addressed

2. Is the manuscript technically sound, and do the data support the conclusions?

Reviewer #1: Yes

Reviewer #2: Yes

3. Has the statistical analysis been performed appropriately and rigorously? 

Reviewer #1: Yes

Reviewer #2: N/A

4. Have the authors made all data underlying the findings in their manuscript fully available?

Reviewer #1: Yes

Reviewer #2: Yes

5. Is the manuscript presented in an intelligible fashion and written in standard English?

Reviewer #1: Yes

Reviewer #2: Yes

6. Review Comments to the Author

Reviewer #1: (No Response)

Reviewer #2: Thank you for addressing my comments.

I do not have any additional comments or concerns about the paper.

Congratulations for the nice paper and , especially, congratulations for the sponsor of this very nice BRAZPD cohort study.

7. PLOS authors have the option to publish the peer review history of their article (what does this mean?). If published, this will include your full peer review and any attached files.

Reviewer #1: No

Reviewer #2: **Yes: **Jose Carolino Divino-Filho

---

## [Editor Report · Acceptance letter]

7 Oct 2021

PONE-D-21-02597R1 

Peritoneal dialysis modality transition and impact on phosphate and potassium serum levels 

Dear Dr. Barretti:

I'm pleased to inform you that your manuscript has been deemed suitable for publication in PLOS ONE. Congratulations! Your manuscript is now with our production department. 

Kind regards, 

on behalf of

Prof Gianpaolo Reboldi 

Academic Editor

PLOS ONE